# Identifying the Most Probable Mammal Reservoir Hosts for Monkeypox Virus Based on Ecological Niche Comparisons

**DOI:** 10.3390/v15030727

**Published:** 2023-03-11

**Authors:** Manon Curaudeau, Camille Besombes, Emmanuel Nakouné, Arnaud Fontanet, Antoine Gessain, Alexandre Hassanin

**Affiliations:** 1Institut de Systématique, Évolution, Biodiversité (ISYEB), Sorbonne Université, MNHN, CNRS, EPHE, UA, 75005 Paris, France; manon.curaudeau1@mnhn.fr; 2Unité Épidémiologie et Physiopathologie des Virus Oncogènes, Institut Pasteur, Université Paris Cité, CNRS UMR 3569, 75015 Paris, France; antoine.gessain@pasteur.fr; 3Unité d’Épidémiologie des Maladies Émergentes, Institut Pasteur, Université Paris Cité, 75015 Paris, France; camille.besombes@pasteur.fr (C.B.); arnaud.fontanet@pasteur.fr (A.F.); 4Department of Arboviruses, Emerging Viruses and Zoonosis, Institut Pasteur, Bangui BP 923, Central African Republic; emmanuel.nakoune@pasteur-bangui.cf; 5Conservatoire National des Arts et Métiers, Unité PACRI, 75003 Paris, France

**Keywords:** Monkeypox, animal reservoir, ecological niche model, tropical Africa, evergreen forests, Sciuridae

## Abstract

Previous human cases or epidemics have suggested that Monkeypox virus (MPXV) can be transmitted through contact with animals of African rainforests. Although MPXV has been identified in many mammal species, most are likely secondary hosts, and the reservoir host has yet to be discovered. In this study, we provide the full list of African mammal genera (and species) in which MPXV was previously detected, and predict the geographic distributions of all species of these genera based on museum specimens and an ecological niche modelling (ENM) method. Then, we reconstruct the ecological niche of MPXV using georeferenced data on animal MPXV sequences and human index cases, and conduct overlap analyses with the ecological niches inferred for 99 mammal species, in order to identify the most probable animal reservoir. Our results show that the MPXV niche covers three African rainforests: the Congo Basin, and Upper and Lower Guinean forests. The four mammal species showing the best niche overlap with MPXV are all arboreal rodents, including three squirrels: *Funisciurus anerythrus*, *Funisciurus pyrropus*, *Heliosciurus rufobrachium*, and *Graphiurus lorraineus*. We conclude that the most probable MPXV reservoir is *F*. *anerythrus* based on two niche overlap metrics, the areas of higher probabilities of occurrence, and available data on MPXV detection.

## 1. Introduction

Monkeypox, now called mpox [1,2], is an emerging zoonotic disease caused by the Monkeypox virus (MPXV). Infection in humans manifests as fever, swollen lymph nodes, and fatigue, followed by a rash with macular lesions progressing to papules, vesicles, pustules, and scabs, usually on the face, hands, and feet for two to five weeks [3].

Taxonomically, MPXV belongs to Poxviridae, a family of large double-stranded DNA viruses (130–375 kbp) represented by 22 genera and 83 species. The family is divided into two subfamilies: Entomopoxvirinae, in which hosts are insects; and Chordopoxvirinae, in which hosts are vertebrates (birds, crocodiles, mammals, and teleost fishes) [4,5,6,7]. Within Chordopoxvirinae, all viruses of the genus *Orthopoxvirus* (OPXV) are related to mammalian hosts, and phylogenetic analyses have supported the existence of an Old World group composed of MPXV and eight other species, such as variola virus (the agent of smallpox), vaccinia virus (the source of modern smallpox vaccines), and cowpox [8]. Mpox was first described in 1958 in Asian macaques used for polio vaccine production and research at the *Statens Serum Institut* in Copenhagen, Denmark [9]. Although the authors concluded that the monkeys were not infected in Denmark, the origin of mpox infection remained elusive. During the following decade, there were several outbreaks in monkeys in laboratories and zoos all over the world: six outbreaks between 1961 and 1966 in macaques and langurs used in American laboratories; two outbreaks between 1964 and 1965 in orangutans, white-handed gibbons, and common marmosets maintained in captivity in Dutch zoos; and one outbreak in 1968 in chimpanzees used in a French laboratory [10]. The first human mpox case was detected in 1970 in a young boy from Bokenda, a remote village in the northwest of the Democratic Republic of Congo (DRC) [11]. In the same year, five cases were identified in West Africa: four in two villages of northeastern Liberia, and one in southern Sierra Leone [12]. Several cases were subsequently discovered in 1971 in Côte d’Ivoire [13] and Nigeria [12], in 1979 in Cameroon [14], in 1983 in the Central African Republic (CAR) [15], and in 1987 in Gabon [16,17]. A total of 404 cases were reported between 1970 and 1986 [18]. The vast majority of these cases originated in remote villages in tropical rainforests (89% of the 283 reported between 1970 and 1984) [18], and most of them were from the DRC.

In 1996–1997, the first large-scale mpox epidemic occurred, with 511 cases in 54 villages of the Katako-Kombe health zone in central DRC [19]. From the 2000s, the number of reported cases has been steadily increasing: from a few hundred cases each year between 2001 and 2009, to more than two thousand cases per year between 2010 and 2014 [20]. Since September 2017, a mpox outbreak has been ongoing in Nigeria [21]. The apparent rise in cases since 2000 could be explained by enhanced field surveillance in African countries [20,22], as well as declining levels of population immunity due to the end of smallpox vaccination, which provided cross-protection against MPXV [23]. Several exports of MPXV have occurred outside the African continent: in 2003 in the United States, in connection with a shipment of wild rodents (*Cricetomys* sp., *Funisciurus* sp., and *Graphiurus* sp.) from Ghana [24,25], in 2018 in Israel [26], in 2018 and 2021 in the United Kingdom [27,28], in 2019 in Singapore [29], and in 2021 in the United States [28], all in connection with travellers to and from Nigeria. The virus has also been exported to South Sudan, a non-endemic African country, in 2005 [30,31]. More recently, more than 80,000 cases have been reported in 2022 worldwide, during an epidemic involving mostly homosexual men [32,33].

Local investigations in African countries have suggested that several human mpox cases or epidemics originated from contacts with wild mammals, such as chimpanzee (*Pan troglodytes*), Pennant’s red colobus (*Piliocolobus pennanti*), or Gambian pouched rat (*Cricetomys emini*) [34,35]. In agreement with that, MPXV has been isolated or sequenced from several mammal species in Africa, including arboreal species, such as monkeys and squirrels, as well as ground-living species, including rodents and shrews [36,37,38,39,40,41]. In addition, antibodies against OPXV were detected in many species from four mammalian orders and nine families. Among these mammal species, one or more could be involved as natural reservoir host(s), in which the virus has been circulating for centuries, while others could be secondary hosts, occasionally contaminated through contact with the reservoir. 

Although the MPXV reservoir has not yet been identified, several lines of evidence point to mammal species endemic to African rainforests of West and Central Africa. First, MPXV belongs to the genus *Orthopoxvirus* (OPXV), a genus exclusive to mammals [5,6]. Second, most human mpox cases have been reported in rainforests of West and Central Africa, or travellers from these regions [24,26,27,28,29,30,31,42]. Third, the geographic distribution of MPXV has been inferred with ecological niche modelling (ENM) methods, and the results have shown that the virus could be found in all rainforests of West and Central Africa [31,43,44,45]. Fourth, phylogenetic studies based on complete MPXV genomes have revealed a strong geographic structure [45,46,47]: viruses from Central Africa (Gabon, Cameroon, CAR, Republic of the Congo, and DRC; clade I [48]) are divergent from those from West Africa; the latter can be separated into two geographic subgroups, one including viruses from Sierra Leone, Liberia, Côte d’Ivoire, and Ghana (clade IIa [48]), and the other including viruses from Nigeria (clade IIb [48]). These results suggest that the animal reservoir populations have been genetically isolated from each other for many generations in three separate rainforest blocks, including the Upper and Lower Guinean forests in West Africa, and the Congo Basin in Central Africa. Therefore, we hypothesized that the MPXV reservoir is represented by one or several rainforest mammal species with a geographic distribution very similar to that of MPXV.

In this study, we explore this hypothesis using a four-step approach: (i) we provide the full list of mammal genera and species previously identified as MPXV natural hosts in Africa; (ii) we predict the geographic distributions (or ecological niche) of all species of these genera based on georeferenced specimens catalogued in museum collection databases and ENM methods; (iii) we reconstruct the ecological niche of MPXV using reliable data on human index cases and MPXV sequences available for georeferenced wild animals; and (iv) we make statistical overlap comparisons between the ecological niches of mammals and MPXV in order to identify the most probable mammalian reservoir(s).

## 2. Materials and Methods

### 2.1. Mammal Species Identified as MPXV Hosts

Many wild mammal species have been linked with a potential MPXV infection in Africa. Some people reported contact with a wild animal prior to an infection (for instance from hunting or through the consumption of bushmeat), while anti-OPXV antibodies have been found in many species [49,50,51,52,53,54,55,56,57]. MPXV was also isolated, and its genome fully sequenced, from two species: one rodent, *Funisciurus anerythrus* (Thomas’s rope squirrel) from Yambuku in the Mongala Province in northern DRC [36,37,38,45], and one primate, *Cercocebus atys* (Sooty Mangabey) from Taï National Park in Côte d’Ivoire [39]. In addition, the MPXV genome was recently sequenced from *P*. *troglodytes* from Taï National Park in Côte d’Ivoire [40], and from six species sampled in DRC, including one shrew (*Crocidura littoralis*) and five rodents, i.e., *Cricetomys* sp., *F*. *anerythrus*, *Funisciurus bayonii*, *Malacomys longipes*, and *Stochomys longicaudatus* [41].

For this study, we focused on all mammal species for which some biological evidence, such as virus isolation, MPXV DNA sequence, PCR amplification, and detection of anti-OPXV antibodies, supports their role as reservoir or secondary hosts. To limit the influence of taxonomic issues at the species level (i.e., no identification, such as *Cricetomys* sp., or possible misidentification), we chose to study every species of all 14 genera of Table 1. Therefore, our taxonomic selection includes 213 African mammal species (full list provided in Appendix A).

### 2.2. Occurrence Records of Selected Mammal Species

To reconstruct the ecological niche of mammal species, we obtained species occurrence data from multiple databases, in an effort to reduce sampling biases [61,62,63]: the Global Biodiversity Information Facility (GBIF; www.gbif.org, accessed on 22 June 2022), Integrated Digitized Biocollections (IdIgBio), VertNet, and INaturalist, using the R package *spocc* [64]. All of these databases carry occurrence records for mammal species, but all have their specificity: GBIF is the largest database of species occurrence records; IdIgBio is the United States digitised collection of specimens [65]; VertNet is a publicly accessible database on vertebrate specimen records from natural history collections around the world [66]; finally, iNaturalist is a citizen-science database with research-grade observations of multiple taxa. Both IdIgBio, VertNet, and Inaturalist communicate part of their data to GBIF, and some records can be found in multiple databases. Occurrence records that were duplicated, erroneous (e.g., in the ocean or in capital cities), or more than 30 km away from the IUCN distribution were filtered out using *CoordinateCleaner* [67]. Finally, occurrence records lower than 10 km apart were filtered using *spThin* [68], as spatial filtering is recommended to improve the performance of ecological niche models and reduce sampling biases [69,70].

### 2.3. Occurrence Records of MPXV Cases

To reconstruct the ecological niche of MPXV, we need occurrence records of MPXV cases. We included the six occurrence records previously published for mammals from which the virus was isolated and/or sequenced, including one *F*. *anerythrus* from Yambuku in the Mongala Province in northern DRC, several *P*. *troglodytes* from the Taï National Park in Côte d’Ivoire, and two primate sanctuaries in Cameroon [36,37,38,40,71]. We also included human index cases, which were presumably infected from an animal source and confirmed by PCR, DNA sequencing, or MPXV isolation. However, the GPS coordinates of mpox cases were not provided in previous ENM studies [31,43,44,45]. All of the 103 human records used in our study are index cases of known village origin [11,12,14,15,16,17,21,34,35,45,46,47,54,59,72,73,74,75,76,77,78,79,80,81,82,83,84,85,86,87,88,89,90], for which the GPS coordinates were recovered using Google Maps (https://www.google.fr/maps, accessed on 22 June 2022), OpenStreetMap (http://www.openstreetmap.org, accessed on 22 June 2022), and Joint Operational Graphic (JOG) topographic maps. Most index cases reported in previous mpox epidemics in Central Africa were young boys living in remote villages surrounded by forests [90,91]. Therefore, we assumed that most human outbreaks began in the forest around the village after direct or indirect contact with an animal infected with MPXV. Since the ecological niche of MPXV was inferred using a cell size of 2.5 min (of latitude) × 2.5 min (of longitude) (see Section 2.4), each of the 103 selected villages was considered to be in the same cell as the nearby forest where the first human infection occurred.

We were able to gather 109 occurrence records for MPXV (Appendix A), including 95 in Central Africa and 14 in West Africa. Occurrence records lower than 10 km apart were filtered out using *spThin* [68], leaving a total of 96 occurrence records, with 84 in Central Africa and 12 in West Africa.

### 2.4. Ecological Niche Modelling

Ecological niche modelling (ENM) correlates occurrence records to environmental variables in order to predict the probability of occurrence for both sampled and non-sampled localities. ENM has been used to infer the distribution of species in the past [92] or future [93], in the context of migration [94], and also to deduce the distribution of invasive species [95] or viruses [96].

Our goal was to reconstruct the ecological niches for MPXV and the 213 selected mammal species. However, the minimum number of occurrence records needed to infer an ecological niche depends on the species prevalence, defined as the fraction of the study area occupied by a species [97]. Here, the species prevalence was calculated as the percentage occupied by the geographic distribution of the species provided by the IUCN [60] on the whole modelling space, i.e., Africa (including Madagascar and other islands). According to van Proodij [98], the minimum number of occurrence records required to build an ecological niche in Africa is 14 for narrow-ranged (with a species prevalence below 0.2), and 25 for widespread species (with a species prevalence above 0.2). 

The ecological niches were reconstructed using occurrence records and variables available in the WorldClim 2.1 [99,100] and ENVIREM [101] datasets at 2.5 min resolution (approximately 4.5 × 4.5 km gridcell). The WorldClim dataset includes 19 bioclimatic variables derived from monthly temperatures and rainfall values; it is the most employed dataset for ENM studies. The ENVIREM (ENVIronmental Rasters for Ecological Modeling) dataset consists of 18 biologically relevant climatic and topographic variables derived from monthly temperature and precipitation data, and monthly extra-terrestrial solar radiation, which are intended to supplement the 19 bioclimatic WorldClim variables [101]. Since one ENVIREM variable (monthCountByTemp10) is categorical, it was not included in the analyses. We added the elevation to these 36 variables and studied them for an area corresponding to a 300 km radius around the selected points, and the *caret* R package [102] was used to determine and select the least correlated variables (|r| < 0.7) [103].

For MPXV and each mammal species, ecological niche modelling was performed with the MaxEnt (Maximum Entropy) algorithm [104,105], a machine learning method, with *ENMTools* in R [106] using 70% of the dataset points as training. To account for the slight differences in results due to the use of a machine learning algorithm [107], the ecological niche modelling was repeated 10 times. The MaxEnt approach was chosen over presence-absence models (Generalized Linear Models (GLM) or BIOCLIM) because it is based on presence-only data, and can produce reliable results even with a limited number of GPS records [108]. The area under the curve (AUC) was used as the measure of model accuracy, with a value of 0.5 indicating model accuracy not better than random, and a value of 1 indicating perfect model fit [109].

To determine if the ecological niche of each mammal species fits well with that of MPXV, niche overlap comparisons were performed with *ENMTools* in R [106], using Schoener’s D [110] and Hellinger’s I [111] metrics. The D and I metrics were obtained by comparing the estimated habitat suitability for each grid cell of the ecological niche models. The closer the D and I values are to 1, the more the niches overlap; in contrary, a value of 0 indicates no overlap between niches [111]. Species were then ranked using their D and I values, and re-ranked according to the average of their ranks.

## 3. Results

### 3.1. Ecological Niche of MPXV

The ecological niche of MPXV was predicted using 96 occurrence records with an AUC of 0.956 (0.946–0.964; SD: 0.005). The best probabilities of occurrence (≥0.25) are indicated by different colours, from blue to yellow, passing through green, in Figure 1A. They delineate a large geographic area that encompasses rainforests of West and Central Africa. In addition, three isolated areas receive some support in East Africa, including one region in Ethiopia, another region to the east of Lake Victoria in Kenya, and another area in the Eastern African coastal forest of Tanzania, around Dar es Salam. In West Africa, the geographic distribution is discontinuous, with a gap in southwestern Togo and Benin. In Central Africa, the niche covers all of the Congo Basin, except a large area in eastern Equatorial Africa, including eastern and southern Gabon, and southern Republic of the Congo. The highest probabilities of occurrence (greater than 0.75, in yellowish green to yellow; Figure 1A) are found in four main regions: (i) eastern Sierra Leone, southern Guinea, and Liberia; (ii) southern Nigeria and adjacent western provinces of Cameroon; (iii) the coasts of southern Cameroon and Equatorial Guinea; and (iv) most parts of the Congo Basin in eastern Republic of the Congo, southern CAR, and DRC.

### 3.2. Ecological Niches of Mammal Species and Overlap with the MPXV Niche

As indicated in Table 1, the 14 genera currently detected as MPXV hosts belong to four mammalian orders, and represent 11 families: Eulipotyphla (Erinaceidae and Soricidae), Macroscelidea (Macroscelididae), Primates (Cercopithecidae, Hominidae, and Lorisidae), and Rodentia (Dipodidae, Gliridae, Muridae, Nesomyidae, and Sciuridae).

In Africa, the 14 genera are represented by 213 species (Appendix A), for which we gathered occurrence records. In the international databases, we found less than 14 occurrence records for 112 species (52.6%) classified as narrow-ranged, and only 17 occurrence records for one species classified as widespread, i.e., *Crocidura viara*. It was therefore impossible to predict the ecological niche of these 113 species. In total, 100 mammal species had enough occurrence data available, after both cleaning and spatial thinning. However, we obtained the ecological niches for only 99 of these 100 species (Figure 1 and Figure 2; electronic Appendix A). It was indeed impossible to infer the niche of *Piliocolobus kirkii* because most occurrence records were found on the edge of environmental variables, and were therefore interpreted as missing data. All of the 99 niches inferred for mammals have an AUC > 0.697, and 79% have an AUC > 0.9 (Appendix A), indicating an excellent performance of the model. Overall, the variability between the 10 replicates of each niche is very low (Appendix A).

Based on niche overlap analyses between MPXV and mammal species, the first ten ranked species are *Funisciurus anerythrus* (Figure 1C), *Graphiurus lorraineus* (Figure 1D), *Funisciurus pyrropus* (Figure 1E), *Heliosciurus rufobrachium* (Figure 1F), *Stochomys longicaudatus* (Figure 2A), *Malacomys longipes* (Figure 2B), *Pan troglodytes* (Figure 2C), *Oenomys hypoxanthus* (Figure 2D), *Crocidura olivieri* (Figure 2E), and *Crocidura theresae* (Figure 2F). The niche overlap with MPXV encompasses both Central and West Africa for all of these species, except *M*. *longipes* and *O*. *hypoxanthus*, which are mainly present in Central Africa, and *C*. *theresae*, which is almost only present in West Africa (Figure 2F).

In general, the ecological niches predicted for mammal species differ from the geographic distributions provided by the IUCN (red lines in Figure 1 and Figure 2; see also electronic Appendix A), either by encompassing more regions (e.g., the niche of *F*. *anerythrus* also contains a large area covering eastern Liberia, Côte d’Ivoire, Ghana, and Togo; Figure 1C) or fewer regions (e.g., the niche of *C*. *olivieri* does not cover the Sahelian savannah zone; Figure 2E), or a combination of both (e.g., the niche of *G*. *lorraineus* also includes the area from Togo to southwestern Nigeria, but does not extend to southeastern DRC; Figure 1D). The niches can also be less fragmented (e.g., the niche of *F*. *pyrropus* is not divided into the four IUCN areas, but rather shows a continuum from Guinea to Uganda and southwestern Kenya; Figure 1E) or more fragmented (e.g., the niche of *O*. *hypoxanthus* shows an Ethiopian patch isolated from the rest of its distribution in Central Africa; Figure 2D).

## 4. Discussion

### 4.1. Ecological Niche of MPXV Fragmented into Three Rainforests

The MPXV niche was previously reconstructed in four studies [31,43,44,45], based on different criteria for the selection of occurrence records: Levine et al. [43] used 156 georeferenced villages in which at least one human case was detected; Lash et al. [44] used 231 occurrence points corresponding to the 404 cases identified by the World Health Organization between 1970 and 1986; Nakazawa et al. [31] completed the dataset with four occurrences from the South Sudan outbreak; and Nakazawa et al. [45] divided the same dataset into 25 subsets of about 41 to 43 localities at least 50 km apart (to account for the higher number of occurrences in DRC). While recent mpox cases were added to our dataset, only 96 occurrence points were selected to infer our niche, as, unlike previous studies [31,43,44,45], we chose to retain only localities of animal cases (6) and human index cases (90). This approach was adopted to focus on human cases with the highest probability of MPXV infection from an animal, and to avoid any bias due to human-to-human transmission. Although based on a different occurrence dataset, our results are quite similar to previous ones, since the five MPXV niches cover, more or less, all rainforests of West and Central Africa. However, it is important to note that our MPXV niche has a fragmented distribution, with at least three (and possibly four) separate rainforest regions (Figure 1B), including in West Africa (i) the Upper Guinean forests (UGF; from Guinea and Sierra Leone through Liberia, Côte d’Ivoire and Ghana to western Togo) and (ii) Lower Guinean forests (LGF; from southeastern Benin through Nigeria to western Cameroon), and in Central Africa (iii) the northern part of Atlantic Equatorial coastal forests (AECF) (from the Sanaga River in southwestern Cameroon to Equatorial Guinea), possibly separated (large area with *p* < 0.5) from (iv) the Congolian lowland forests (CLF; from southeastern Cameroon and northeastern Gabon through northern Republic of the Congo and CAR to DRC). Such a fragmentation of the niche fits well with previous results [31], but contrasts with several niches showing a division into only two areas, corresponding to UGF, and LGF, AECF, and CLF [43,44,45].

In agreement with our MPXV niche divided into three or four rainforests, phylogeographic studies on MPXV [45,46,47] have provided strong support for a sister-group relationship between UGF and LGF viruses, a lineage uniting the viruses from Cameroon and Gabon (AECF), and its grouping with several CLF virus lineages. Taken together, these results suggest that important biogeographic barriers have prevented or limited the dispersal of the MPXV mammalian reservoir in only two regions: (i) between UGF and LGF, where two different barriers may have been involved, the Volta River in eastern Ghana and the Dahomey Gap, a savannah corridor that extends from eastern Ghana through Togo and Benin [112]; and (ii) between LGF and the Congo Basin, where the barrier is obviously the Sanaga River, the largest river in Cameroon. Interestingly, these natural barriers are the distribution limits for a wide diversity of arboreal mammal species [60], including several primates, squirrels, and genets: the Volta River is the eastern limit for the green monkey (*Chlorocebus sabaeus*), two squirrels (*Heliosciurus punctatus* and *Protoxerus aubinnii*), and the pardine genet (*Genetta pardina*), and the western limit for the large-spotted genet (*Genetta maculata*); the Dahomey Gap is the eastern limit for the spot-nosed monkey (*Cercopithecus petaurista*) and *G*. *pardina*; the Sanaga River is the northern limit for five primates (*Arctocebus aureus*, *Colobus satanas*, *Euoticus elegantulus*, *Mandrillus sphinx*, *Sciurocheirus gabonensis*), the ribboned rope squirrel (*Funisciurus lemniscatus*), and the servaline genet (*Genetta servalina*); and the Sanaga River is the southern limit for five primates (*Arctocebus calabarensis*, *Cercopithecus erythrotis*, *Cercopithecus mona*, *Euoticus pallidus*, *Sciurocheirus alleni*) and the crested genet (*Genetta cristata*). Several phylogeographic studies, based on DNA sequences, have shown that these barriers can also constitute boundaries between mammal subspecies [113,114,115]. The most convincing study concerns anthropoid apes, as whole genome data analyses [113] have suggested that from the Middle/Late Pleistocene to present, the Sanaga River has separated two subspecies of chimpanzee (*Pan troglodytes troglodytes* and *P*. *t*. *ellioti*) and two subspecies of lowland gorilla (*Gorilla gorilla gorilla* and *G*. *g*. *diehli*), and that from the Middle Pleistocene until today, the Volta River and Dahomey Gap have isolated the western chimpanzee (*P*. *t*. *verus*) from the Nigeria-Cameroon chimpanzee (*P*. *t*. *ellioti*).

### 4.2. Which Mammal Species Are the Most Probable Reservoir Hosts?

Of the 213 species belonging to the 14 mammalian genera potentially involved as reservoir or secondary hosts of MPXV, 113 did not have sufficient occurrences to reconstruct an ecological niche (< 14 for narrow-ranged species and < 25 for widespread species). Among these 113 species, none are distributed in both West and Central Africa, indicating that they cannot be considered as likely reservoir host species for MPXV. Among the 99 mammal species for which it was possible to reconstruct a niche, 49 were only distributed in either West African rainforests (28) or Central African rainforests (21), and 22 were found in both West and Central African rainforests (Appendix A). Finally, only seven mammalian niches cover, with high probabilities (greater than 0.75, highlighted in yellowish green to yellow in Figure 1 and Figure 2), the three rainforests corresponding to the Upper Guinean forests (UGF), Lower Guinean forests (LGF), and the Congo Basin, and they were logically found among the top ten classified mammalian niches for their overlap with the MPXV niche. They are represented by three squirrels (family Sciuridae) of the tribe Protoxerini, i.e., *Funisciurus anerythrus* (Thomas’s rope squirrel; rank one; Figure 1C), *Funisciurus pyrropus* (red-legged rope squirrel; rank three; Figure 1E), and *Heliosciurus rufobrachium* (red-legged sun squirrel; rank four; Figure 1F), one arboreal rodent of the family Gliridae, *Graphiurus lorraineus* (Lorrain dormouse; rank two; Figure 1D), one ground-living rodent of the family Muridae, *Stochomys longicaudatus* (target rat; rank five; Figure 2A), *Pan troglodytes* (chimpanzee; family Hominidae; rank seven; Figure 2C), and *Crocidura olivieri* (African giant shrew; family Soricidae; rank nine; Figure 2E). MPXV was detected directly (PCR and DNA sequencing) and/or indirectly (anti-OPXV antibodies) in all of these species, except *C*. *olivieri* (Table 1).

All seven niches, except that of *H*. *rufobrachium*, show significant differences with the IUCN range maps. First, some large IUCN areas are missing in several niches, such as northeastern CAR in the niche of *F*. *anerythrus*, Gabon and northern Angola in the niche of *F*. *pyrropus*, and the Sahelian savannah zone in the niche of *C*. *olivieri*. Since the gaps are always in peripheral distribution, the first hypothesis is to consider that the species is not present, or rarely, in these areas. As an alternative, we can propose that the IUCN range maps were drawn using some specimens misidentified at the species level. This hypothesis can hold for species of the genera *Crocidura*, *Funisciurus*, and *Graphiurus*, as they contain several closely related species that are phenotypically very similar and are therefore concerned by taxonomic issues [60,116]. Second, several niches show important extensions by comparison with the IUCN range maps, either eastward (e.g., *P*. *troglodytes*: in Uganda, southwestern Kenya, and northwestern Tanzania), westward (e.g., *F*. *anerythrus* and *S*. *longicaudatus*: from Liberia through Côte d’Ivoire and Ghana to Togo), or linking separate geographical areas (e.g., *F*. *pyrropus*: the three northern IUCN areas form a continuum supported by high probabilities). Although high probabilities of occurrence indicate favourable environmental conditions for these species, they can be effectively absent in these suitable areas due to significant biogeographic barriers, such as large rivers and savannah corridors, or disadvantageous competition with one or more previously established species. In agreement with this view, we hypothesize that the current absence of *P*. *troglodytes* in East Africa (Uganda, Kenya, Tanzania, and also Ethiopia) is the consequence of competition with other anthropoid species during the Pliocene and Pleistocene epochs. This is corroborated by the discovery of many hominid fossils in East Africa [117], and by the recent disappearance of chimpanzee from southwestern Kenya, where they were still present in the Middle Pleistocene [118]. For poorly known taxa, such as many rodents and shrews, high probabilities of occurrence may indicate that these species are effectively present in these suitable areas, but they have not been considered present by the IUCN due to species misidentifications. This hypothesis can be advanced for the two similar species of *Funisciurus*, *F*. *anerythrus* and *F*. *pyrropus*, for which the niches were found to be very different from the IUCN maps. In particular, the niche predicted for *F*. *anerythrus* suggests that it could be present in the UGF, whereas its IUCN range map shows that *F*. *anerythrus* does not occur west of the Dahomey gap (Figure 1C). Taxonomically, *F*. *anerythrus* has often been confused with *F*. *pyrropus*: *F*. *anerythrus* was first described as a subspecies of *F*. *pyrropus* [116], and one subspecies of *F*. *pyrropus* from Gambia in West Africa, i.e., *F*. *p*. *mandingo*, was placed in *F*. *anerythrus* by some taxonomists [119]. Another species, *Funisciurus substriatus* from northeastern Ghana, southeastern Burkina Faso, Togo, and Benin, may be also conspecific with *F*. *anerythrus* [116]. To address these taxonomic issues, *Funisciurus* specimens from museum mammal collections should be examined for morphology (e.g., pelage colour, pattern of stripes, body measurements, skull characteristics) and sequenced for mitochondrial and nuclear genes.

## 5. Conclusions

Our MPXV niche and recent phylogeographic studies on MPXV [45,46,47] suggest that two biogeographic barriers have isolated mammalian populations of the MPXV reservoir host: on the one hand, the Volta River and Dahomey Gap between UGF and LGF, and on the other hand, the Sanaga River between LGF and the Congo Basin. Our analyses revealed that the niche of *F*. *anerythrus* shows the best overlap (using both D and I metrics) with that of MPXV, suggesting that the Thomas’s rope squirrel could be the main MPXV reservoir. Interestingly, *F*. *anerythrus* is the only species from which MPXV was isolated [36,37] and fully sequenced by two independent teams in different provinces of northern DRC [41,45]. In addition, two small fragments of MPXV were amplified in museum specimens from five *Funisciurus* species, all collected in rainforests of the Congo Basin: *F*. *anerythrus* (45 out of 362 specimens tested; 12,4%), *F*. *carruthersi* (3 of out 109 specimens tested; 2.8%), *F*. *congicus* (32 out of 239 specimens tested; 13.4%), *F*. *lemniscatus* (5 of out 82 specimens tested; 6.1%), and *F*. *pyrropus* (8 of out 201 specimens tested; 4.0%) [58]. These data suggest that the MPXV reservoir could contain not just one species, but several species of *Funisciurus*. In the Congo Basin, however, *F*. *anerythrus* is sympatric with the four other species, including *F*. *pyrropus* and the three species endemic to Central Africa, *F*. *carruthersi*, *F*. *congicus*, and *F*. *lemniscatus*. As a consequence, it can be hypothesized that *F*. *anerythrus* is indeed the reservoir host species, which can frequently contaminate other arboreal species, such as the squirrels and monkeys listed in Table 1 (secondary hosts), due to regular contacts (direct or indirect) in forest trees. Two complementary studies need to be conducted to further investigate this hypothesis: (i) *Funisciurus* squirrels caught in future field surveys in African forests and those currently housed in museum mammal collections should be systematically tested for the presence of MPXV; and (ii) they should be sequenced for mitochondrial and nuclear genes to compare the phylogeography of *F*. *anerythrus* with that already available for MPXV.

## Figures and Tables

**Figure 1 viruses-15-00727-f001:**
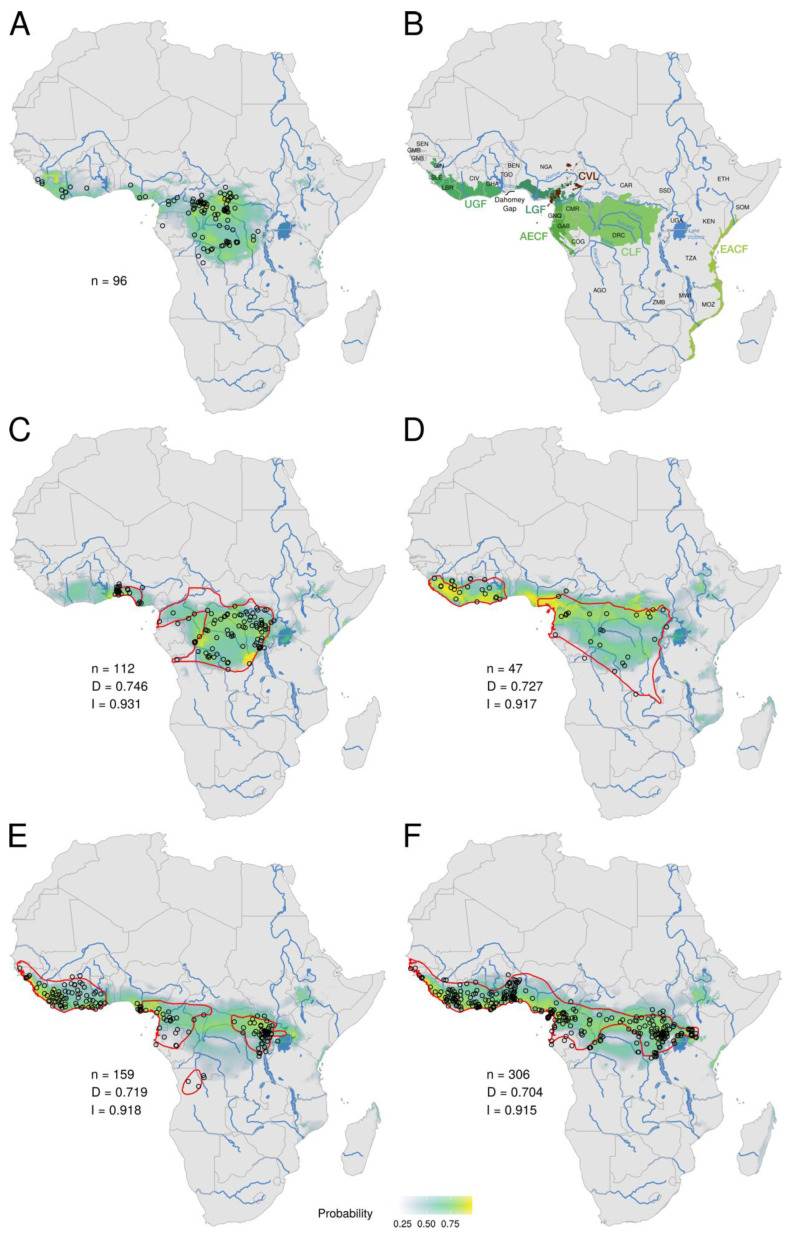
Ecological niches of Monkeypox virus (**A**) and the four mammal species showing the best overlap with it: *Funisciurus anerythrus* (**C**), *Graphiurus lorraineus* (**D**), *Funisciurus pyrropus* (**E**), and *Heliosciurus rufobrachium* (**F**). Black circles indicate localities used to build the distribution model. The probabilities of occurrence (*p*) are highlighted using different colours: blue grey for probabilities < 0.5; turquoise green for 0.5 < *p* < 0.75; yellowish green for 0.75 < *p* < 0.9; and yellow for *p* > 0.9. The red line is the IUCN distribution of the species [60]. Indicated at the left of the maps are the number of occurrence records (n) used to infer the ecological niche, and the Schoener’s D and Hellinger’s I values summarizing niche overlap between mammal species and MPXV. For convenience, we have included a map (**B**) showing the major biogeographic barriers, such as the Dahomey gap, rivers, and Cameroon volcanic line (CVL), and African rainforests (**B**), including the Upper Guinean forests (UGF) and Lower Guinean forests (LGF) in West Africa, the Atlantic Equatorial coastal forests (AECF) and Congolian lowland forests (CLF) in Central Africa, and the Eastern African coastal forests (EACF) in East Africa.

**Figure 2 viruses-15-00727-f002:**
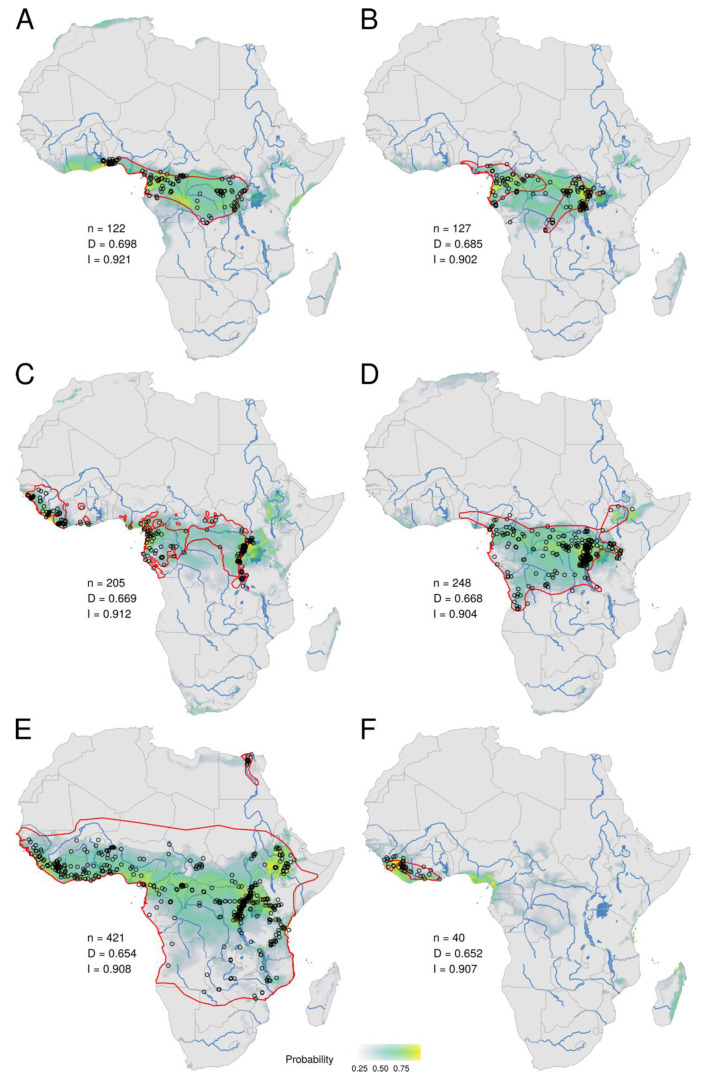
Ecological niches of mammal species ranked between the 5th and 10th positions for their overlap with the MPXV niche. *Stochomys longicaudatus* (**A**), *Malacomys longipes* (**B**), *Pan troglodytes* (**C**), *Oenomys hypoxanthus* (**D**), *Crocidura olivieri* (**E**), and *Crocidura theresae* (**F**). See legend of Figure 1 for more details.

**Table 1 viruses-15-00727-t001:** Mammal genera and species potentially involved as reservoir or secondary hosts of MPXV.

Order	Family	Genus	Virus Isolation	MPXV Fragment Amplified by PCR (Sequenced in Bold)	Anti-OPXV Antibodies
Eulipotyphla	Erinaceidae	Atelerix		*Atelerix* spp. [55]	*Atelerix* spp. [55]
Eulipotyphla	Soricidae	Crocidura		***C. littoralis*** [41]	
Macroscelidea	Macroscelididae	Petrodromus			*P. tetradactylus* [54,57]
Primates	Cercopithecidae	Allenopithecus			*A. nigroviridis* [52]
Primates	Cercopithecidae	Cercocebus	*C*. *atys* [39]	***C*. *atys*** [39]	*C. galeritus* [52]
Primates	Cercopithecidae	Cercopithecus			*C. ascanius* [38,51,52,53] *C. mona* [52] *C. nictitans* [52] *C. petaurista* [49,50] *C. pogonias* [38,51,52]
Primates	Cercopithecidae	Chlorocebus			*C. aethiops* * [49]
Primates	Cercopithecidae	Piliocolobus			*P. badius ** [50] *P. pennanti ** [52]
Primates	Hominidae	Pan		***P. troglodytes*** [40]	
Primates	Lorisidae	Perodicticus			*P. potto* [52]
Rodentia	Dipodidae	Jaculus		*Jaculus* spp. [55]	
Rodentia	Gliridae	Graphiurus		*G. lorraineus* [55] *Graphiurus* spp. [55,56]	*G. lorraineus* [57] *Graphiurus* spp. [55,56]
Rodentia	Muridae	Malacomys		***M. longipes*** (MT724769)	
Rodentia	Muridae	Oenomys			*O. hypoxanthus* [57]
Rodentia	Muridae	Stochomys		***S. longicaudatus*** [41]	
Rodentia	Nesomyidae	Cricetomys		***Cricetomys* sp. [41]** *Cricetomys* spp. [55,56]	*C. emini* [54,57] *Cricetomys* spp. [55,56]
Rodentia	Sciuridae	Funisciurus	*F. anerythrus* [37]	***F. anerythrus*** [37,41,55,56,58] ***F. bayonii*** [41] *F. carruthersi* [58] *F. congicus* [58] *F. lemniscatus* [58] *F. pyrropus* [58] *Funisciurus.* spp. [55,56]	*F. anerythrus* [38,51,52,54] *F. isabella* [52] *F. lemniscatus* [52] *Funisciurus* spp. [52,53,56,57,59]
Rodentia	Sciuridae	Heliosciurus			*H. rufobrachium* [51,52,53,54] *H. gambianus* [52,56] *Heliosciurus* spp. [39,53,57]
Rodentia	Sciuridae	Xerus		*Xerus* sp. [56]	

*: current species names provided by the IUCN [60].

## Data Availability

The data presented in this study are available in Appendix A.

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
