# Peer review of "Identifying the Most Probable Mammal Reservoir Hosts for Monkeypox Virus Based on Ecological Niche Comparisons"

_viruses, 2023, doi:10.3390/v15030727_

Round 1
Reviewer 1 Report
MPXV Review
Thank you for the opportunity to read this generally well designed and well referenced study using ecological niche modelling of pathogen distribution and potential host distribution to propose several hosts of mpox that require further investigation. This work is useful for generating hypotheses around which species may be hosts but there are some significant limitations to these approaches that are not highlighted in this manuscript. I feel that these should be discussed in greater depth within this work than they are currently. At a minimum I think that confidence intervals around the predicted ranges should be presented and that there should be a discussion of the effect of the underlying biases in both mpox occurrence points and species sampling across their ranges. If these are addressed I feel that this piece of work could be of assistance to the scientific community in thinking about how to investigate potential hosts of known and novel zoonoses.
I have made several other minor suggestions and expand on the two main comments below.
Change Monkeypox infection to mpox throughout as appropriate, as per recent WHO guidance.
Define secondary hosts, versus detection of incidental infection.
Introduction:
L34-35; change to “infection in humans manifests as fever…”
L36; may be worth referencing some of the articles reviewing acute cases from the current outbreak which suggests that secondary human-to-human transmission may result in different symptomatology. As this article focusses on spillover from zoonotic transmission this might be worth pointing out.
L62-66; may be worth mentioning that the apparent rise in increase of infections could be due to several factors, first, increased case finding and diagnostics and second, the waning levels of population immunity to the indirect protection from smallpox vaccination.
Methods:
L136-149: It is good that the authors have combined data from 4 sources of information to identify occurrence records. However, a recent review that I wrote specifically investigate the spatial biases that are typically present within these datasets that should be mentioned as a limitation of these approaches. We previously found that low proportions of a species known range provides most of these occurrence points and this data is even more skewed for rarely detected species which will be the case for rodents that are primarily arboreal. (https://journals.plos.org/plosntds/article?id=10.1371/journal.pntd.0010772). I would suggest that at the least these biases are acknowledged. It may be beneficial to use data available from scientific and grey literature sources (i.e. conservation reports) to supplement the currently used data. I understand that this may add considerable effort at this stage to incorporate these additional data sources and so this is purely a suggestion and should not prevent the current analysis being reported as conducted.
I have another question about the use of cleaning based on IUCN distributions. I have observed that some of these are not too accurate for some species and so would the authors be able to share the impact of this step. How many occurrence points were discarded and did this step affect some species more than others?
L152-168: The assumption that all cases are presumable infected from an animal source is quite strong. More so that the location of a case is equivalent to the location at which the case was infected. It is however, a necessary assumption for the purposes of this study and so I think this should be discussed further, either in the methods section or in the discussion.
Again, the bias in these samples are quite apparent. It may be of interest to the readers to compare the number of cases for which you have occurrence data with the number of cases that have been reported from the region over the same time frame. It would seem to me that location data from the many cases that have been reported from Nigeria are not available which may importantly bias the ENM if current data are only clustered in the Lower Guinean Forest region and the Congo basin. Have you performed any sensitivity analysis when using state or district level locations from Nigeria or other country data?
Another potential bias to consider is the temporal component. Landuse, and population change across West Africa has been substantial in the 50 years from which cases arise. Is it possible to consider the impact that this may have on your models?
L184-194: How did you handle the translation of your point data to the raster data from the environmental variables. What cell sizes did you use? Why did you not use any landuse variables in your model, it seems that you heavily relied on environmental variables but if your hypothesis was that arboreal and forest dwelling mammals where the primary sources of spillover it seems this would have been a useful variable to include?
L206-212: Do these metrics weight the overlap by the probability of occurrence for both of the outputs you are comparing. i.e. are cells where the probability of occurrence is 1 and 1 weighted more than where they may be 1 and 0.25 respecitively?
Figure 1: A. mpox ENM is it possible to share the partial dependence plots for these models particularly for the mpox virus which others are being compared too. It would be instructive to understand which of your variables are explaining the greatest impact on probability of occurrence. The model does not seem to predict well for some cases, with most having <50% probability of occurrence at the location of the case. Is it possible to provide complementary figures to 1A in the supplementary displaying the uncertainty around this prediction estimate? Figure 1B needs a description in the figure legend.
Results:
L234-249: Alongside the comments in Fig1A above it would be informative to have some understanding on the variables that describe this distribution. What were the coefficients for these models? What are the confidence intervals around these predictions. See (https://besjournals.onlinelibrary.wiley.com/doi/full/10.1111/2041-210X.13389) for a useful discussion on the benefits of describing confidence around the central estimate from SDMs.
L276-287: Perhaps given that you removed occurrence sites outside of the IUCN range your results may be an argument for keeping them in if you are showing high probability of occurrence outside of the range. If you would not expect that it perhaps suggests that the models are not accurately predicting the species ranges.
Discussion:
L348-352: This seems a strong statement. Why does the reservoir for MPXV need to be the same species in its entire range? Could it not have a different but closely related species in each of its divided regions and if so this would go against your point here.
I think it would be beneficial to expand the discussion with some of the points I have raised above. The focus on IUCN misattribution for species ranges while important does not describe all of the potential affects of biases in the underlying data on your presented distribution.
Kind regards,
Author Response
Dear Editor,
Please find below our answers to the comments made by the reviewer#1.
Reviewer#1
Thank you for the opportunity to read this generally well designed and well referenced study using ecological niche modelling of pathogen distribution and potential host distribution to propose several hosts of mpox that require further investigation. This work is useful for generating hypotheses around which species may be hosts but there are some significant limitations to these approaches that are not highlighted in this manuscript. I feel that these should be discussed in greater depth within this work than they are currently. At a minimum I think that confidence intervals around the predicted ranges should be presented and that there should be a discussion of the effect of the underlying biases in both mpox occurrence points and species sampling across their ranges. If these are addressed I feel that this piece of work could be of assistance to the scientific community in thinking about how to investigate potential hosts of known and novel zoonoses.
Thank you for your comments on our work.
I have made several other minor suggestions and expand on the two main comments below.
Change Monkeypox infection to mpox throughout as appropriate, as per recent WHO guidance.
Changed as requested by reviewer#1.
The first sentence has been changed to “Monkeypox (MPX), now called mpox is an emerging zoonotic disease caused by the Monkeypox virus (MPXV).”
Define secondary hosts, versus detection of incidental infection.
We have already defined secondary hosts in the text as ‘occasionally contaminated through contact with the reservoir’ (line 87).
Introduction:
L34-35; change to “infection in humans manifests as fever…”
Changed as requested by reviewer#1.
L36; may be worth referencing some of the articles reviewing acute cases from the current outbreak which suggests that secondary human-to-human transmission may result in different symptomatology. As this article focusses on spillover from zoonotic transmission this might be worth pointing out.
We do not agree with the suggestion made by reviewer#1.
Our introduction was written chronologically and we tried to summarize all known cases of zoonotic transmissions and human epidemics. So, mentioning the different symptoms related to the current epidemic seems out of place.
L62-66; may be worth mentioning that the apparent rise in increase of infections could be due to several factors, first, increased case finding and diagnostics and second, the waning levels of population immunity to the indirect protection from smallpox vaccination.
Improved as requested by reviewer#1. We added the following sentence (lines 66-69): ‘The apparent rise in cases since 2000 could be explained by enhanced field surveillance in African countries [20,22], as well as declining levels of population immunity due to the end of smallpox vaccination which provided cross-protection against MPXV [23].’
Methods:
L136-149: It is good that the authors have combined data from 4 sources of information to identify occurrence records. However, a recent review that I wrote specifically investigate the spatial biases that are typically present within these datasets that should be mentioned as a limitation of these approaches. We previously found that low proportions of a species known range provides most of these occurrence points and this data is even more skewed for rarely detected species which will be the case for rodents that are primarily arboreal. (https://journals.plos.org/plosntds/article?id=10.1371/journal.pntd.0010772). I would suggest that at the least these biases are acknowledged. It may be beneficial to use data available from scientific and grey literature sources (i.e. conservation reports) to supplement the currently used data. I understand that this may add considerable effort at this stage to incorporate these additional data sources and so this is purely a suggestion and should not prevent the current analysis being reported as conducted.
We agree with Reviewer#1 that international databases could contain some biased or erroneous information for ‘rarely detected species’. That’s why we used four rather than only one database.
We are aware that the number of occurrences for each species could be raised by using grey literature. However, this would require a tremendous effort both for to collect and standardise data considering we analysed 213 different species.
We changed the text as follows in lines 140-143:
‘To reconstruct the ecological niche of mammal species, we obtained species occurrence data from multiple databases, in an effort to reduce sampling biases [61–63]: the Global Biodiversity Information Facility (GBIF; www.gbif.org), Integrated Digitized Biocollections (IdIgBio), VertNet and Inaturalist using the R package spocc [64].’
and in line 152-155:
‘Finally, occurrence records lower than 10 km apart were filtered using spThin [68], as spatial filtering is recommended to improve the performance of ecological niche models and reduce sampling biases [69,70].’
Furthermore, rarely detected mammal species are not so problematic for our study because it was not possible to infer their ecological niche. Indeed, we wrote in lines 257-260 that ‘In the international databases, we found less than 14 occurrence records for 112 species (52.6%) classified as narrow-ranged, and only 17 occurrence records for one species classified as widespread, i.e., Crocidura viara. It was therefore impossible to predict the ecological niche of these 113 species.’
I have another question about the use of cleaning based on IUCN distributions. I have observed that some of these are not too accurate for some species and so would the authors be able to share the impact of this step. How many occurrence points were discarded and did this step affect some species more than others?
For information, the number of discarded occurrence points was provided in Table S1 for all mammalian species.
Indeed, some IUCN distribution are not the most accurate in some cases. However, we needed a common ground for all species.
Cleaning was not only done based on IUCN distributions but (as indicated in lines 150-155) ‘Occurrence records that were duplicated, erroneous (e.g., in the ocean or in capital cities), or more than 30 km away from the IUCN distribution were filtered out using CoordinateCleaner [67]. Finally, occurrence records lower than 10 km apart were filtered using spThin [68], as spatial filtering is recommended to improve the performance of ecological niche models and reduce sampling biases [69,70]’.
L152-168: The assumption that all cases are presumable infected from an animal source is quite strong. More so that the location of a case is equivalent to the location at which the case was infected. It is however, a necessary assumption for the purposes of this study and so I think this should be discussed further, either in the methods section or in the discussion.
As already mentioned in our manuscript, the ecological niche of the MPVX was reconstructed in previous studies using all available human cases. In our study, we selected only animal cases and index cases of human epidemics. As stated in the discussion (lines 306-308) ‘This approach was adopted to focus on human cases with the highest probability of MPXV infection from an animal and to avoid any bias due to human-to-human transmission.’
In order to highlight the difference with previous studies, we did not discriminate between index cases and human to human transmission. We transformed the sentence in the discussion (lines 304-307) as follows:
‘While recent mpox cases were added to our dataset, only 96 occurrence points were selected to infer our niche, as, unlike previous studies [31,43,44,45], we chose to retain only localities of animal cases (6) and human index cases (90).’
Again, the bias in these samples are quite apparent. It may be of interest to the readers to compare the number of cases for which you have occurrence data with the number of cases that have been reported from the region over the same time frame. It would seem to me that location data from the many cases that have been reported from Nigeria are not available which may importantly bias the ENM if current data are only clustered in the Lower Guinean Forest region and the Congo basin. Have you performed any sensitivity analysis when using state or district level locations from Nigeria or other country data?
We do not agree with reviewer#1 because we focused on index cases with a probable zoonotic transmission. In Nigeria, most of the cases are resulting from human to human transmission and were therefore not considered. In addition, we pooled all cases, not taking into account the date, because the number of index cases was too small to consider making different ENMs for different time frame.
Another potential bias to consider is the temporal component. Landuse, and population change across West Africa has been substantial in the 50 years from which cases arise. Is it possible to consider the impact that this may have on your models?
Although we agree with reviewer#1 that landuse and population changes probably had an impact on the dynamics of epidemics in West Africa, this is not the subject of our present study. Our study concerns the identification of the animal reservoir by comparing the ecological niche of the MPXV with the ecological niches of 99 mammal species. This bias would have been the same for both the virus and the animal species. It must be noted that the climatic parameters used for our study include monthly temperature (minimum, maximum and average), precipitation, solar radiation, vapour pressure and wind speed, aggregated across a target temporal range of 1970–2000, using data from between 9000 and 60 000 weather stations (https://doi.org/10.1002/joc.5086). In this context, it is not possible to consider the temporal component in our study.
L184-194: How did you handle the translation of your point data to the raster data from the environmental variables. What cell sizes did you use? Why did you not use any landuse variables in your model, it seems that you heavily relied on environmental variables but if your hypothesis was that arboreal and forest dwelling mammals where the primary sources of spillover it seems this would have been a useful variable to include?
We chose to use environmental rasters that were openly and readily available for the entire African continent. We used 2.5 min of a degree for cell sizes. Since our occurrence points were located 10 km one from another, they were not on the same cells.
As explained above, landuse and population changes are not the subject of our present study.
L206-212: Do these metrics weight the overlap by the probability of occurrence for both of the outputs you are comparing. i.e. are cells where the probability of occurrence is 1 and 1 weighted more than where they may be 1 and 0.25 respecitively?
Yes, we confirm that these metrics weight the overlap by the probability of occurrence for both of the outputs (see Warren et al. 2008; doi: 10.1111/j.1558-5646.2008.00482.x).
Figure 1: A. mpox ENM is it possible to share the partial dependence plots for these models particularly for the mpox virus which others are being compared too. It would be instructive to understand which of your variables are explaining the greatest impact on probability of occurrence. The model does not seem to predict well for some cases, with most having <50% probability of occurrence at the location of the case. Is it possible to provide complementary figures to 1A in the supplementary displaying the uncertainty around this prediction estimate? Figure 1B needs a description in the figure legend.
As requested by Reviewer#1, we have improved the legend of Figure 1B as follows (lines 244-250):
‘For convenience, we have included a map (B) showing the major biogeographic barriers, such as the Dahomey gap, rivers and Cameroon volcanic line (CVL), and African rainforests (B), including the Upper Guinean forests (UGF) and Lower Guinean forests (LGF) in West Africa, the Atlantic Equatorial coastal forests (AECF) and Congolian lowland forests (CLF) in Central Africa, and the Eastern African Coastal Forests (EACF) in East Africa. and the major biogeographic barriers, such as the Dahomey gap, rivers and Cameroon volcanic line (CVL).’
Results:
L234-249: Alongside the comments in Fig1A above it would be informative to have some understanding on the variables that describe this distribution. What were the coefficients for these models? What are the confidence intervals around these predictions. See (https://besjournals.onlinelibrary.wiley.com/doi/full/10.1111/2041-210X.13389) for a useful discussion on the benefits of describing confidence around the central estimate from SDMs.
We agree with reviewer#1 that all methods can be improved. However, this is not the purpose of our study to test new methods for which robustness still remains uncertain. Confidence intervals discussed in this article cited by Reviewer#1 paper are for species distribution modelling computed with Bayesian additive regression trees, which is not the case in our study, and are therefore not applicable.
As indicated in Materials and Methods (lines 200-209), ‘For MPXV and each mammal species, ecological niche modelling was performed with the MaxEnt (Maximum Entropy) algorithm [103,104], a machine learning method, with ENMTools in R [105] using 70% of the dataset points as training. To account for the slight differences of results due to the use of a machine learning algorithm [106], the ecological niche modelling was repeated 10 times. The MaxEnt approach was chosen over presence-absence models (Generalized Linear Models (GLM) or BIOCLIM) because it is based on presence-only data and can produce reliable results even with a limited number of GPS records [107]. The area under the curve (AUC) was used as the measure of model accuracy, a value of 0.5 indicating model accuracy not better than random, and a value of 1 indicating perfect model fit [108].’
We used 10 repetitions to account for the slight differences of results due to the use of a machine learning algorithm as recommended by Sillero et al. 2021 (doi: 10.1080/13658816.2020.1798968). We gave the minimum and maximum AUC for each model in Table S1.
The article describing MaxEnt is currently cited 16,792 times, which is much more than the 43 citations of the method cited by reviewer#1.
L276-287: Perhaps given that you removed occurrence sites outside of the IUCN range your results may be an argument for keeping them in if you are showing high probability of occurrence outside of the range. If you would not expect that it perhaps suggests that the models are not accurately predicting the species ranges.
Because we considered 213 species, we had to have a standard protocol for all species. For some species, data points outside of the IUCN range could indeed belong to the species, but this would require manual verification and we could not do this for every specimens. That’s why we used a buffer of ‘30 km [around] the IUCN distribution’ to account for such points.
Discussion:
L348-352: This seems a strong statement. Why does the reservoir for MPXV need to be the same species in its entire range? Could it not have a different but closely related species in each of its divided regions and if so this would go against your point here.
This point is currently discussed in lines 425-439 as follows:
‘These data suggest the MPXV reservoir could contain not just one species but several species of Funisciurus. In the Congo Basin, however, F. anerythrus is sympatric with the four other species, including F. pyrropus and the three species endemic to Central Africa, F. carruthersi, F. congicus, and F. lemniscatus. As a consequence, it can be hypothesized that F. anerythrus is indeed the reservoir host species, which can contaminate frequently other arboreal species, such as squirrels and monkeys listed in table 1 (secondary hosts), due to regular contacts (direct or indirect) in forest trees. Two complementary studies need to be conducted to further investigate this hypothesis: (i) Funisciurus squirrels caught in future field surveys in African forests and those currently housed in museum’s mammal collections should be systematically tested for the presence of MPXV; and (ii) they should be examined for morphology (e.g., pelage colour, pattern of stripes, body measurements, skull characters) and sequenced for mitochondrial and nuclear genes to solve some taxonomic problems involving F. anerythrus, F. pyrropus and F. substriatus and to compare the phylogeography of F. anerythrus with that already available for MPXV.’
I think it would be beneficial to expand the discussion with some of the points I have raised above. The focus on IUCN misattribution for species ranges while important does not describe all of the potential affects of biases in the underlying data on your presented distribution.
Kind regards,’
Reviewer 2 Report
This article is well-written to provide prediction of the most probable natural reservoir of MPXV. Therefore, I only have some minor comments:
Minor comments:
- In the introduction section, the authors need to mention the recent classification of MPXV (clade 1, 2 and 3 based on the analysis of the recent outbreaks).
- Line 153-154: Since the authors only focused in confirmed MPXV (PCR, sequencing, MPXV isolation), there was a possibility that they will miss many other MPXV cases. Have the authors discussed it in limitations of the study? How can it influence prediction?
- Line 300: How do the authors select MPXV cases with the highest probability from the reported MPXV cases and those cases were not due to human-to-human transmission?
- Table 1, the title for table has yet to be written.
- In the conclusion section, the authors need to discuss in one last paragraph, what is the implications of their findings? Do we need to enhance our surveillance in that species? Or performed serological studies to see the seroprevalence? Other potential implications can be discussed.
Author Response
Dear Editor,
Please find below our answers to the comments made by the reviewer#2.
Reviewer#2
This article is well-written to provide prediction of the most probable natural reservoir of MPXV. Therefore, I only have some minor comments:
Thank you greatly for this feedback on our work.
Minor comments:
- In the introduction section, the authors need to mention the recent classification of MPXV (clade 1, 2 and 3 based on the analysis of the recent outbreaks).
We have added the classification in the introduction section (lines 95-100) as follows:
‘Fourth, phylogenetic studies based on complete MPXV genomes have revealed a strong geographic structure [45–47]: viruses from Central Africa (Gabon, Cameroon, CAR, Republic of Congo, and DRC; clade I [48]) are divergent from those from West Africa; and the latter can be separated into two geographic subgroups, one including viruses from Sierra Leone, Liberia, Côte d’Ivoire and Ghana (clade IIa [48]), and the other including viruses from Nigeria (clade IIb [48]).’
- Line 153-154: Since the authors only focused in confirmed MPXV (PCR, sequencing, MPXV isolation), there was a possibility that they will miss many other MPXV cases. Have the authors discussed it in limitations of the study? How can it influence prediction?
While it is true that by focusing on only confirmed MPXV, some real cases may have been excluded, we preferred to have fewer but high quality points for niche reconstruction.
- Line 300: How do the authors select MPXV cases with the highest probability from the reported MPXV cases and those cases were not due to human-to-human transmission?
Unlike previous studies that did not really discriminate between index cases and transmission between humans, we only used index cases and animal cases. Indeed, previous studies controlled for human transmission within the same village by keeping only one point per village. However, some cases were the results of interhuman transmission between villages. In our study, these cases were not considered.
- Table 1, the title for table has yet to be written.
Thank you for pointing this out. The title of the table ‘Mammal genera and species potentially involved as reservoir or secondary hosts of MPXV’ has been added.
- In the conclusion section, the authors need to discuss in one last paragraph, what is the implications of their findings? Do we need to enhance our surveillance in that species? Or performed serological studies to see the seroprevalence? Other potential implications can be discussed.
Thank you for this suggestion. A sentence has been added at the end of the paragraph (lines 432-439) as follows:
‘Two complementary studies need to be conducted to further investigate this hypothesis: (i) Funisciurus squirrels caught in future field surveys in African forests and those currently housed in museum’s mammal collections should be systematically tested for the presence of MPXV; and (ii) they should be examined for morphology (e.g., pelage colour, pattern of stripes, body measurements, skull characters) and sequenced for mitochondrial and nuclear genes to solve some taxonomic problems involving F. anerythrus, F. pyrropus and F. substriatus and to compare the phylogeography of F. anerythrus with that already available for MPXV.’
Round 2
Reviewer 1 Report
I am pleased to see that the authors have responded to some of the comments I raised at the previous round of revisions. However, I still feel that further improvement could be made to the manuscript. I have limited these comments to those that could hopefully be addressed with minimal additional effort but that would contribute to a better understanding of the study as the authors present it.
In my previous comment I requested the authors to acknowledge the taxonomic and spatial biases inherent in the underlying data sources for this work. The additional comments they have added to the text do not adequately do this, although they are important additions for other reasons. Combining the 4 datasets as the authors have done so will improve the coverage of data and the number of contributing data points. However, unless the sampling processes and study designs that have led to the generation of that data are importantly different and designed to reduce these biases, combining these four will not reduce the sampling biases. In fact, if the same sampling biases are present in all four datasets the combination of them will instead exacerbate the issue of sampling bias when using this data. This is the data that is available to the researchers, and I understand that it is not feasible to estimate the degree of bias for each of the species from each of the datasets but I would still strongly recommend the authors to incorporate a discussion of this point in the limitations section of the manuscript.
Previously I mentioned that there is an assumption that the location of identification of a case is equivalent to the location of infection of a case. I do not think this has been adequately addressed in the revised manuscript. I believe this is an important assumption. The incubation period is up to 17 days and humans that have been infected elsewhere may only be detected as cases when they return to their home village. You are therefore designating the point of infection (to investigate overlap with potential host ranges) as a different location to the “true“ location of infection. It may be that individuals in this endemic region only move within short distances of their home locations and that the environmental conditions in all of the areas they occupy are the same but I do not think you can implicitly make this assumption without mentioning it in the manuscript.
In my comment on the results section I requested that the authors provide any estimate on the confidence of their estimates from their produced models. They do not seem to have addressed this point. I personally do not use MaxENT and so had shared an article that discusses presentation of confidence around point estimates more generally in species distribution modelling. However, I have identified the following articles that do discuss confidence intervals from a MaxENT perspective (https://www.biorxiv.org/content/10.1101/603100v1.full, https://onlinelibrary.wiley.com/doi/full/10.1111/j.1600-0587.2013.07872.x, https://biodiversityinformatics.amnh.org/open_source/maxent/Maxent_tutorial2017.pdf). I would be surprised if the authors were not able to generate any meaningful measure of the uncertainty around their proposed point estimates of the probability of occurrence of species from a tool that has been cited 16,792 times. The authors use overlap of a host and case distribution model to infer whether the host is responsible for being the reservoir of the pathogen. If the probability of occurrence were to a wide range of possible values across the region you could potentially get very different results for your likely species. Because of this I believe it is important to communicate your uncertainty through confidence intervals or sensitivity analysis. Neither of which is currently done.
If these additional three comments are addressed I think this manuscript would be more robust and would be of greater use to researchers trying to understand the risk of spillover of mpox in endemic settings.
Author Response
Dear editor,
Please find below our answers (in green) to the comments made by reviewer#1.
I am pleased to see that the authors have responded to some of the comments I raised at the previous round of revisions. However, I still feel that further improvement could be made to the manuscript. I have limited these comments to those that could hopefully be addressed with minimal additional effort but that would contribute to a better understanding of the study as the authors present it.
Thank you again for your comments on our work.
In my previous comment I requested the authors to acknowledge the taxonomic and spatial biases inherent in the underlying data sources for this work. The additional comments they have added to the text do not adequately do this, although they are important additions for other reasons. Combining the 4 datasets as the authors have done so will improve the coverage of data and the number of contributing data points. However, unless the sampling processes and study designs that have led to the generation of that data are importantly different and designed to reduce these biases, combining these four will not reduce the sampling biases. In fact, if the same sampling biases are present in all four datasets the combination of them will instead exacerbate the issue of sampling bias when using this data. This is the data that is available to the researchers, and I understand that it is not feasible to estimate the degree of bias for each of the species from each of the datasets but I would still strongly recommend the authors to incorporate a discussion of this point in the limitations section of the manuscript.
We agree with reviewer#1 that spatial and taxonomic biases are inherent in any study based on occurrences records. This is precisely the reason why we decided to reconstruct ecological niches with occurrences records extracted from four different databases rather than using geographic ranges provided by the IUCN. We also indicated in lines 152-155 that ‘occurrence records lower than 10 km apart were filtered using spThin [68], as spatial filtering is recommended to improve the performance of ecological niche models and reduce sampling biases [69,70].’ In addition, spatial and taxonomic biases were discussed in our manuscript in section 4.2 (lines 399-435). In particular, we identified a strong spatial bias due to a possible taxonomic problem between F. anerythrus and F. pyrropus in West Africa. To clarify, we added the following sentence to the text (lines 435-438): ‘To address these taxonomic issues, Funisciurus specimens from museum’s mammal collections should be examined for morphology (e.g., pelage colour, pattern of stripes, body measurements, skull characters) and sequenced for mitochondrial and nuclear genes.’
Previously I mentioned that there is an assumption that the location of identification of a case is equivalent to the location of infection of a case. I do not think this has been adequately addressed in the revised manuscript. I believe this is an important assumption. The incubation period is up to 17 days and humans that have been infected elsewhere may only be detected as cases when they return to their home village. You are therefore designating the point of infection (to investigate overlap with potential host ranges) as a different location to the “true“ location of infection. It may be that individuals in this endemic region only move within short distances of their home locations and that the environmental conditions in all of the areas they occupy are the same but I do not think you can implicitly make this assumption without mentioning it in the manuscript.
We agree with reviewer#1 that we can never be totally sure of the true location of infection.
However, as previously stated, we only focused on index cases. Unlike secondary cases, who for example may have been infected in hospitals (see reference [77]), these cases are considered to have been infected near their village. To better address this point, section 2.2 has been rewritten as follows:
‘To reconstruct the ecological niche of MPXV, we need occurrence records of MPXV cases. We included the six occurrence records previously published for mammals from which the virus was isolated and/or sequenced, including one F. anerythrus from Yambuku in the Mongala Province in northern DRC and several P. troglodytes from the Taï National Park in Côte d’Ivoire and two Primate sanctuaries in Cameroon [36–38,40,71]. We also included human index cases, which were presumably infected from an animal source and confirmed by PCR, DNA sequencing, or MPXV isolation. However, the GPS coordinates of mpox cases were not provided in previous ENM studies [31,43–45]. All the 103 human records used in our study are index cases of known village origin [11,12,14–17,21,34,35,45–47,54,59,72–90], for which the GPS coordinates were recovered using Google Maps (https://www.google.fr/maps), OpenStreetMap (http://www.openstreetmap.org), and Joint Operational Graphic (JOG) topographic maps. Most index cases reported in previous mpox epidemics in Central Africa were young boys living in remote villages surrounded by forests [90,91]. We therefore assumed that most human outbreaks began in the forest around the village after direct or indirect contact with an animal infected with MPXV. Since the ecological niche of MPXV was inferred using a cell size of 2.5 min (of latitude) x 2.5 min (of longitude) (see section 2.4), each of the 103 selected villages was considered to be in the same cell than the nearby forest where the first human infection occurred.’
In my comment on the results section I requested that the authors provide any estimate on the confidence of their estimates from their produced models. They do not seem to have addressed this point. I personally do not use MaxENT and so had shared an article that discusses presentation of confidence around point estimates more generally in species distribution modelling. However, I have identified the following articles that do discuss confidence intervals from a MaxENT perspective (https://www.biorxiv.org/content/10.1101/603100v1.full, https://onlinelibrary.wiley.com/doi/full/10.1111/j.1600-0587.2013.07872.x, https://biodiversityinformatics.amnh.org/open_source/maxent/Maxent_tutorial2017.pdf). I would be surprised if the authors were not able to generate any meaningful measure of the uncertainty around their proposed point estimates of the probability of occurrence of species from a tool that has been cited 16,792 times. The authors use overlap of a host and case distribution model to infer whether the host is responsible for being the reservoir of the pathogen. If the probability of occurrence were to a wide range of possible values across the region you could potentially get very different results for your likely species. Because of this I believe it is important to communicate your uncertainty through confidence intervals or sensitivity analysis. Neither of which is currently done.
As stated in section 2.4, each ecological niche was inferred using 10 replicates in order to take into account possible variability (lines 201-203): ‘To account for the slight differences of results due to the use of a machine learning algorithm [106], the ecological niche modelling was repeated 10 times.’
We also indicated the average, minimum and maximum of the 10 AUCs in Table S1.
However, we have decided to follow the recommendation of reviewer#1 and two additional results were added in our manuscript:
1. The 95% confidence intervals for the AUCs were added in Table S1.
2; A new figure was included in the supplementary as supplementary materials. Figure S2: Standard deviation calculated for the ecological niche of MPXV and that of the 10 mammal species showing the best overlap with the MPXV niche. It shows that there is little variability between the 10 replicates for the virus and the top 10 species. In addition, each of the 10 models for the tested species were compared in the overlap analyses to each of the 10 MPXV models, reducing the possible bias due to variability. The following sentence was added to the text: ‘Overall, the variability between the 10 replicates of each niche is very low (Figure S2)’ in the text (lines 265-266).
If these additional three comments are addressed I think this manuscript would be more robust and would be of greater use to researchers trying to understand the risk of spillover of mpox in endemic settings.